# The Role of Artificial Intelligence in Predicting the Progression of Intraocular Hypertension to Glaucoma

**DOI:** 10.3390/life15060865

**Published:** 2025-05-27

**Authors:** Nicoleta Anton, Cătălin Lisa, Bogdan Doroftei, Ruxandra Angela Pîrvulescu, Ramona Ileana Barac, Ionuț Iulian Lungu, Camelia Margareta Bogdănici

**Affiliations:** 1Department of Ophtalmology, “Grigore T. Popa” University of Medicine and Pharmacy, 16 Universității Street, 700115 Iasi, Romania; anton.nicoleta1@umfiasi.ro (N.A.); camelia.bogdanici@umfiasi.ro (C.M.B.); 2Ophthalmology Clinic, Sf. Spiridon Emergency Clinical Hospital, 700111 Iasi, Romania; 3Department of Chemical Engineering, Faculty of Chemical Engineering and Environmental Protection 11 Cristofor Simionescu, Gheorghe Asachi Technical University of Iasi, 73, Prof.dr.doc. D. Mangeron Street, 700050 Iasi, Romania; catalin.lisa@academic.tuiasi.ro; 4Department of Mother and Child Care, “Grigore T. Popa” University of Medicine and Pharmacy, 700115 Iasi, Romania; bogdandoroftei@gmail.com; 5Department of Ophtalmology, “Carol Davila” University of Medicine and Pharmacy, 050474 Bucharest, Romania; ramona.barac@umfcd.ro; 6Faculty of Pharmacy, “Grigore T. Popa” University of Medicine and Pharmacy, 16 Universității Street, 700115 Iasi, Romania; ionut-iulian.lungu@umfiasi.ro

**Keywords:** artificial intelligence, glaucoma, intraocular hypertension, artificial neural networks

## Abstract

AI systems, especially artificial neural networks (ANNs), are increasingly involved in the diagnosis and personalized management of ophthalmologic disorders. Background: This study shows the practical applications of artificial intelligence for predicting the progression of intraocular hypertension (IOH) to glaucoma. Methods: This study involved two groups of patients with IOH and a control group, analyzed using the commercial Neurosolution simulator. The findings were compared with experimental data. The performance of the neural models was evaluated using several metrics: Mean Squared Error (MSE), Normalized Mean Squared Error (NMSE), correlation coefficient (r^2^), and percentage error (Ep). Results: For all three patient groups, the best performance was achieved with neural networks featuring two hidden layers: MLP(9:18:9:3) for group 1, MLP(10:20:10:3) for group 2, and MLP(10:30:20:3) for group 3. The MSE values during validation were 0.39 for groups 1 and 2, and 0.34 for group 3. For these neural networks, the probability of producing correct outputs during validation was 75% (i.e., 9 correct responses out of a possible 12). The findings in this study are in line with those reported by other researchers in the field. Conclusions: The neural network models developed in this study demonstrated their potential for predicting the progression of intraocular hypertension to glaucoma.

## 1. Introduction

Glaucomatous optic neuropathy is the leading cause of irreversible blindness worldwide. Glaucoma is a progressive, neurodegenerative optic neuropathy of multifactorial etiology that results in the death of retinal ganglion cells (RGCs). Glaucomas comprise a group of diseases that lead to the progressive loss of RGCs, producing a characteristic pattern of optic nerve head damage and visual field (VF) defects. According to studies conducted on healthy individuals, the association between glaucoma and intraocular pressure (IOP) was historically considered absolute—patients with an IOP more than two standard deviations above the population mean (often above 21 mmHg) were believed to inevitably develop glaucoma.

However, modern clinical, histopathological, and experimental evidence suggests that the optic nerve head is the primary site of IOP-induced injury in glaucoma. Glaucomatous damage is caused by a range of risk factors, including elevated intraocular pressure, ischemic perfusion injury, structural abnormalities of the lamina cribrosa, and suspected intrinsic factors acting at the level of RGCs and their supporting glia—such as oxidative stress, calcium dysregulation, and increased intracellular signaling cascades.

IOP is known to exert pathological effects by inducing biomechanical stress at the level of the lamina cribrosa and by disrupting axoplasmic flow, indirectly impairing RGC axonal perfusion—either individually or through a combination of these mechanisms and others. Another recognized mechanism is hypoperfusion of the optic nerve head, implicating systemic blood pressure as a contributing factor in glaucoma pathogenesis. [1]

Diagnosing and monitoring the disease require combining clinical examination findings with subjective data from visual field testing and objective biometric measurements such as pachymetry, corneal hysteresis, and imaging of the optic nerve and retina. In clinical practice, glaucoma is typically diagnosed and followed using a multi-modal approach, including intraocular pressure (IOP) measurement through tonometry, visual field tests, optical coherence tomography (OCT), and fundoscopic examination [2,3,4]

Each of these diagnostic tools has its limitations: tonometry readings can be influenced by corneal thickness, visual field tests depend on the patient’s attention and cooperation, and OCT and fundus exams require expert interpretation, often with a high degree of subjectivity [2,3]. Furthermore, patients with intraocular hypertension who are at risk of developing glaucoma need to be monitored regularly to allow for early detection and timely intervention. Artificial intelligence (AI) has emerged as a valuable tool for harnessing this extensive data, offering automated, consistent, and predictive support across all stages of glaucoma care [3,4].

Neural networks have been widely applied in medical diagnostics due to their ability to generalize—meaning they can operate with data different from the ones used during training once the models have been validated [5,6,7]. Several studies in the literature have demonstrated the potential of artificial intelligence (AI) in ophthalmology as well as in other branches of medicine. In oncology, neural networks integrated with CT, MRI, and PET imaging have been proposed for the detection of brain tumors. More recently, AI has been used to predict medical events and patient outcomes [8,9,10]. Studies have shown that neural networks can estimate survival rates in patients with breast, colorectal [10,11], lung [12], and prostate cancer [13] more accurately than traditional clinical assessments. In obstetrics and gynecology, AI tools are also gaining ground. The ability of AI to process and store large volumes of data can support the identification of risk factors for premature labor. Moreover, in assisted reproduction, AI is being used to select the most viable oocytes and embryos [14,15,16]. Artificial neural networks can be useful tools for the prediction of several neurodevelopmental outcomes, and their predictive performance can be improved by including a large number of clinical and paraclinical parameters [17].

A recent study suggested that incorporating machine learning algorithms into first-trimester screening for preeclampsia (PE) and intrauterine growth restriction (IUGR) could enhance the overall detection rate of these conditions. However, this hypothesis should be validated in larger groups of pregnant women from various geographic regions [18].

Artificial intelligence tools, and especially artificial neural networks, are progressively involved in detecting and customizing the control of ophthalmic diseases. In ophthalmology, artificial intelligence is applied in several areas, most commonly in the diagnosis and prediction of conditions such as glaucoma, diabetic retinopathy, age-related macular degeneration, and retinopathy of prematurity. Particularly, glaucoma has been extensively studied, with several studies supporting the use of AI tools to predict disease progression through image analysis [3,5]. The precise data of the explorations and, particularly, the emergence of new imaging methods (OCT) have led to the increase in interest in the use of these tools by multidisciplinary teams. The combination of AI technologies and optical coherence tomography (OCT) proved to be trustworthy in detecting retinopathies or in enhancing the diagnostic conduct of retinal diseases [5,6,7]. In a study we carried out in 2021, the combination of DE and SVM proved to be effective, with the methodology offering relevant results for the current issue: an accuracy of 100% for the training set and 95.23% for the test set, with only one sample being incorrectly rated. The study was conducted on a sample of 52 patients: particularly, 101 eyes with glaucoma and diabetes mellitus, in the Ophthalmology Clinic I of the “St. Spiridon” Clinical Hospital of Iasi. The criteria considered in the modeling action were normal or hypertensive open-angle glaucoma, intraocular hypertension, and associated diabetes [19]. In another recent work (2022), various machine learning algorithms aiming at estimating the progression of open-angle glaucoma (POAG) were used. The evaluation of glaucoma progression was conducted based on parameters such as VFI (Visual field index), MD (Mean Deviation), PSD (Pattern Standard Deviation), and RNFL (retinal nerve fiber layer). The best results of over 90% accuracy were achieved by Multilayer Perceptron and Random Forest algorithms [20]. Additional studies further support the link between sleep apnea syndrome and glaucomatous changes. In this study, it was found that the increased rate of sleep apnea syndrome produced a severe ocular surface disorder and a neurodegenerative disorder of the retina. The eyes of patients with sleep apnea syndrome (SAS) and glaucoma had lower mean intraocular pressure than the eyes with glaucoma without SAS. However, the mean C/D ratio in eyes with glaucoma correlated with the severity of SAS. Applied neural network models have demonstrated their potential in predicting glaucoma progression in patients with coexisting sleep apnea. During the validation phase, most of the calculated parameters fell within a ±25% confidence interval, thus reinforcing the connection between sleep apnea syndrome and glaucoma-related changes, as previously reported in the literature [21,22]. More recent studies and meta-analyses (8, 2025), including 48 studies, have demonstrated that deep learning (DL) algorithms exhibit high diagnostic performance in detecting glaucoma using fundus photography and OCT imaging. This can be achieved by recognizing subtle structural changes indicative of glaucoma, such as the optic nerve head morphology (e.g., the cup-to-disc ratio) or the thickness of the retinal nerve fiber layer. Additionally, by incorporating clinical data and visual field measurements, DL models can extract spatiotemporal features that may provide improved assessments of glaucoma progression [23]. Another recent review of studies in the literature shows that in glaucoma, AI can help analyze large amounts of data from diagnostic tools, such as fundus images, optical coherence tomography, and visual field tests [24].

In light of recent results and the critical importance of early glaucoma diagnosis—particularly in preventing disease progression—this study aimed to employ artificial intelligence tools to predict the progression of intraocular hypertension to glaucoma.

## 2. Materials and Methods

### 2.1. Study Group

To develop the neural networks, we used NeuroSolutions 4.01, a specialized software product created by NeuroDimension, Inc. (Gainesville, FL, USA). This simulator is based on visual programming and provides users with predefined neuron models, data interaction modules, and training algorithm components. These elements are easy to configure and visually represented, enabling the intuitive construction of neural network structures. NeuroSolutions also offers continuous and direct control over neural network parameters—even during the training process—making it a versatile tool for designing and managing artificial neural networks. Through its graphical interface, users can combine modular blocks to generate a wide range of neural architectures. The type of neural network used is the Multilayer Perceptron (MLP) with feedforward error propagation.

The Multilayer Perceptron is a feedforward neural network with one or more hidden layers, consisting of the following:An input layer;One or more hidden layers;An output layer;Computation occurs only in the hidden and output layers;Input signals are propagated forward through each layer of the network.

A hidden layer is referred to as such because its desired output is not explicitly known; given the input–output mapping of the entire network (treated as a “black box”), it is not possible to determine the expected output of individual neurons in hidden layers. Commercial neural networks typically have one or two hidden layers, each containing between 10 and 1000 neurons. Experimental networks may include 3 or 4 hidden layers with millions of neurons.

The number of layers—and more importantly, the number of neurons in each layer—is generally determined through trial and error, with the goal of achieving optimal model performance.

In this study, feedforward neural networks (Multilayer Perceptrons) were developed using two hidden layers for each of three patient groups. The constructed database included data from the following groups:

Group 1: 75 patients with untreated intraocular hypertension (IOH);

Group 2: 70 patients with treated IOH;

Group 3 (control group): 85 patients with primary open-angle glaucoma (POAG).

A total of nine input parameters were considered: age, sex, months since diagnosis, diagnosis group, systolic blood pressure, diastolic blood pressure, pachymetry, maximum intraocular pressure (IOP max), and minimum intraocular pressure (IOPmin). The model outputs consisted of three parameters: PSD (Pattern Standard Deviation), RISC (risk of glaucoma progression), and C/D ratio (cup-to-disc ratio).

The number of neurons per layer varied between 9 and 30. Neural network training was performed using NeuroSolutions, a dedicated software developed by NeuroDimension. The TanhAxon activation function was used in all layers. The training process, which depended on the number of epochs, lasted less than 8 min in all cases. The Momentum learning algorithm was employed.

The optimal number of training epochs was determined by evaluating the model’s performance as the number of epochs increased. For networks trained on Group 1 and Group 3, the optimal value was 80,000 epochs, while for Group 2, the optimal number was 60,000 epochs.

The most important stage in the neural network modeling was validation, during which a dataset that was not used during training was employed to evaluate the model’s ability to generalize (Figure 1).

The criteria used to select the optimal network topology included the following performance metrics:Mean Squared Error (MSE);Correlation coefficient (r^2^);Percentage error (Ep%).

The topology of each network was encoded as (m:n:*p*), where m represents the number of neurons in the input layer; n represents the number of neurons in the hidden layer(s); and *p* represents the number of neurons in the output layer.

The most critical phase in the modeling process was validation, where the model’s responses were evaluated using data that were not included in the training set, in order to assess the network’s generalization ability.

The performance of the neural network models was evaluated by calculating the Mean Squared Error (MSE), Normalized Mean Squared Error (NMSE), correlation coefficient (r^2^), and percentage error (Ep). The closer the MSE was to zero and the r^2^ value was to 1, the better the model’s performance.

### 2.2. Statistical Analysis

The database was created in the EXCEL program and was statistically processed with the SPSS 18.0 program. The ANOVA (analysis of variance) test, Student’s *t* test, χ^2^ test, Kruskal–Wallis test, non-parametric test comparing 3 or more groups, the correlation between different phenomena performed using the “r” (Pearson) correlation coefficient, the receiver operating characteristic (ROC) curve highlighting the specificity/sensitivity balance as a prognostic factor, and logistic regression (multivariate analysis) were used as analytical tests

## 3. Results

### 3.1. Statistical Analysis Results

Prior to modeling with artificial neural networks (ANNs), statistical processing of the dataset was conducted using SPSS version 18.0.

The study group consisted of 234 patients, divided into three subgroups based on the management of intraocular hypertension (IOHT) and glaucoma risk, as follows:

Group I—75 patients with untreated IOHT;

Group II—70 patients with treated IOHT;

Group III—89 patients with treated primary open-angle glaucoma (POAG)—control group.

Sex distribution showed a higher proportion of female patients (77.8%), with no statistically significant differences between the study groups (*p* = 0.505).

Age distribution:

The patients’ ages ranged from 50 to 86 years, with an overall mean age of 59.38 ± 11.66 years. Across the three study groups, age distribution was statistically homogeneous (*p* = 0.353), although the mean age was slightly higher in the treated IOH group (61.07 years) compared to the untreated IOH group (58.61 years) and the treated POAG group (58.71 years) (Figure 2).

The age group distribution revealed the following statistically significant findings (*p* = 0.004):

The highest frequency of patients with untreated intraocular hypertension (IOH) was observed in the 50–59 age group (40%) and the 60–69 age group (36%).

Among the patients with treated IOH, the peak frequency was in the 60–69 age group (42.9%).

For patients with treated primary open-angle glaucoma (POAG), the highest frequency was also in the 60–69 age group (51.7%).

Systolic blood pressure ranged from 93.33 to 171 mmHg, with a significantly higher mean value recorded in the treated IOH group (132.10 mmHg, *p* = 0.011).

Diastolic blood pressure ranged from 93.33 to 110 mmHg, with a slightly higher average observed in the untreated IOH group (84.43 mmHg), though this difference was not statistically significant (*p* = 0.622).

Central corneal thickness (pachymetry) showed no significant differences in mean values between the three groups (*p* = 0.375).

The mean intraocular pressure was significantly lower in the treated POAG group (see Table 1).

The mean Pattern Standard Deviation (PSD) was significantly lower in patients with treated IOH (*p* = 0.003).

The 5-year risk of conversion from intraocular hypertension (IOHT) to glaucoma was significantly higher in untreated IOHT patients (16.83%) compared to those with treated IOHT (12.21%) and treated POAG (control group) (11.24%), with a statistically significant difference (*p* = 0.001) (Table 2).

The conversion risk from untreated intraocular hypertension (IOH) to glaucoma was found to be nearly twice as high as in treated patients (RR = 1.98; 95% CI: 1.05–3.15; *p* = 0.002). These results demonstrate that the 5-year risk of glaucoma development doubles in the absence of treatment, and after 7 years, the risk becomes approximately three times higher (see Figure 3).

Linear regression model 7 showed that in 59% of cases, individual values of blood pressure, central corneal thickness, intraocular pressure, pattern deviation, and the cup-to-disc ratio were significant predictors of an increased risk score for glaucoma progression (*p* = 0.001) (Table 3).

### 3.2. Neural Network Modeling Result

Before making the prediction with neural networks, we analyzed the database. Based on the analysis of the database, where we identified statistically significant correlations, we included input and output parameters in this study. Another important landmark was linear regression model 7, which showed that in 59% of cases, individual values of blood pressure, central corneal thickness, intraocular pressure, pattern deviation, and the cup-to-disc ratio were significant predictors of an increased risk score for glaucoma progression (*p* = 0.001). The 5-year risk of conversion from intraocular hypertension (IOHT) to glaucoma was significantly higher in untreated IOHT patients compared to those with treated IOHT.

Thus, a total of nine input parameters were considered: age, sex, months, diagnosis group (DG), systolic blood pressure, diastolic blood pressure, pachymetry, maximum intraocular pressure (IOP max), and minimum intraocular pressure (IOP min). The three output parameters were as follows: Pattern Standard Deviation (PSD), risk score (RISC), and cup-to-disc ratio (C/D). These parameters were used in all three groups.

Thus, we tried to obtain the best networks for each group that would make the best predictions about treated IOP in glaucoma versus untreated IOP. The presence of intraocular hypertension treatment may positively influence the delay of intraocular hypertension progression in glaucoma.

#### 3.2.1. The First Dataset Consisted of 75 Entries, Which Were Randomly Divided into 63 Entries for the Training Stage and 12 for the Testing Stage

To randomly select the data used for the training and testing stages, the database was processed as follows: each data series was assigned a random number using the Excel function INT(row)·number of data + 1, then the data series were sorted in ascending order based on the assigned number. Various types of transfer functions and training algorithms were tested. The best results were found using the Tanh Axon transfer function and the Momentum training algorithm. The Momentum training algorithm was used with a learning rate (α) of 0.01 and a Momentum coefficient (β) of 0.9. As shown in Figure 4, the optimal number of training iterations was 80,000.

A total of nine input parameters were considered: age, sex, months, diagnosis group (DG), systolic blood pressure, diastolic blood pressure, pachymetry, maximum intraocular pressure (IOP max), and minimum intraocular pressure (IOP min). The three output parameters were as follows: Pattern Standard Deviation (PSD), risk score (RISC), and cup-to-disc ratio (C/D).

The Kruskal–Wallis one-way analysis of variance on ranks revealed statistically significant differences in the median values among the treatment groups (H = 849.848, degrees of freedom = 11, *p* < 0.001), indicating that the observed differences were unlikely to have occurred by chance. To compare the datasets, we performed the Student–Newman–Keuls (SNK) test, which was chosen to identify the specific differences between groups following the significant Kruskal–Wallis test. The pairwise comparisons for the experimental data resulted in statistically significant differences (*p* < 0.05). This confirmed the presence of substantial differences between all evaluated groups, consistent with the non-parametric nature of the data.

Table 4 shows the topologies of the Multilayer Perceptron (MLP) neural networks constructed for the first dataset, along with their performance metrics, including Mean Squared Error (MSE), Normalized Mean Squared Error (NMSE), correlation coefficient (r^2^), and percentage error (Ep).(1)$$MSE=\frac{\sum_{j=1}^{P}\sum_{i=1}^{N}{\left(d_{ij}-y_{ij}\right)}^{2}}{N\cdot P}$$
where *p* represents the number of output variables (in this case, *p* = 1), *N* is the number of data points, is the output value for element *i* after processing element *j*, and is the desired (target) output for element *i* after processing element *j*;(2)$$NMSE=\frac{\bar{{\left(Y_{exp}-Y_{net}\right)}^{2}}}{\bar{Y_{exp}}⸱\bar{Y_{net}}}$$(3)$$r^{2}=\frac{\sum \left(Y_{\exp_{i}}-{\overset{-}{Y}}_{\exp}\right)\cdot \left(Y_{net_{i}}-{\overset{-}{Y}}_{net}\right)}{\sqrt{{\sum {\left(Y_{\exp_{i}}-{\overset{-}{Y}}_{\exp}\right)}^{2}\cdot \left(Y_{net_{i}}-{\overset{-}{Y}}_{net}\right)}^{2}}}$$(4)$$E_{p}=\frac{Y_{\exp}-Y_{net}}{Y_{\exp}}\cdot 100$$
where *Y* represents the output data values, with exp and *net* denoting the experimental values and those retrieved from the neural network models, respectively. The correlation coefficient (r^2^) was calculated using Equation (3). This coefficient can range from −1 to +1, where a value of 0 shows no correlation, +1 indicates a perfect positive correlation, and −1 shows a perfect negative correlation.

The network topology was encoded as MLP (m:n:*p*), where m represents the number of neurons in the input layer, n is the number of neurons in the hidden layer, and *p* is the number of neurons in the output layer (as shown in Table 4).

According to the results shown in Table 4, the best performance is achieved by the MLP (9:18:9:3) model. For this model, Figure 5 and Figure 6 compare the experimental values with those predicted by the model during the training and validation phases.

In the training phase, very good correlation coefficients, with values close to 1, were observed, confirming that the neural model accurately correlated the output parameters with the input parameters. According to the results shown in Figure 6, the neural model had a 75% probability of producing correct responses during the validation phase (specifically, 9 responses close to the experimental values out of a possible 12).

Testing of the best-performing neural model for this group of patients with untreated ocular hypertension was performed by predicting PSD, RISK, and C/D for a set of five patients (Figure 7). After obtaining these predictions, PSD, RISK, and C/D were evaluated in these patients, and it was found that in only two out of five cases (40%) were there significant differences between experimental values and neural model predictions for PSD and C/D, and in one out of five cases (20%) for RISK.

#### 3.2.2. The Second Dataset of Patients with Treated Ocular Hypertension Comprises 70 Datasets

Of these, 58 were randomly selected for use in the training stage, while the remaining data were reserved for the validation stage. To evaluate the differences between the inputs and outputs of the database, we first performed the Shapiro–Wilk test for normality, which indicated a failure to meet the normality assumption (*p* < 0.05). As a result, we proceeded with the non-parametric Kruskal–Wallis one-way analysis of variance on ranks, which revealed statistically significant differences in the median values among the treatment groups (H = 865.380, degrees of freedom = 12, *p* < 0.001). This suggested that the observed differences were unlikely to be due to chance. Additionally, to compare the datasets, we also performed the Student–Newman–Keuls (SNK) test for this database. Statistically significant differences were obtained (*p* < 0.05), indicating substantial differences between all evaluated groups, consistent with the non-parametric nature of the data.

Several Multilayer Perceptron (MLP) neural network models with feedforward error propagation were constructed for this dataset, with their performances shown in Table 5. According to the results shown in Figure 8, the optimal number of training epochs determined for this dataset was 60,000. The neural models’ performances shown in Table 5 correspond to this number of training epochs.

Figure 9 presents a comparative view of the experimental values and those predicted by the best-performing MLP model (10:20:10:3). Correlation coefficients of 1 were obtained for RISK and C/D, and 0.9999 for PSD.

The results shown in Figure 10 indicate the probability of obtaining correct responses from this neural model during the validation stage. This probability was 67% (8 responses close to experimental values out of 12) for PSD, 83% (10 responses close to experimental values out of 12) for RISK, and 75% (9 responses close to experimental values out of 12) for C/D.

#### 3.2.3. The Third Database (Control Group) Contains 89 Datasets Randomly Divided into 77 for Training and 12 for Validation

Applying the non-parametric one-way Kruskal–Wallis analysis of variance on ranks to this dataset, we found statistically significant differences in the median values among the treatment groups (H = 1103.741, degrees of freedom = 12, *p* < 0.001). Furthermore, the Student–Newman–Keuls (SNK) test indicated statistically significant differences (*p* < 0.05) between all evaluated groups, consistent with the non-parametric nature of the data.

The performances of the neural models developed for this database are shown in Table 6, corresponding to an optimal number of training epochs of 80,000, as indicated by the results shown in Figure 11.

In Figure 12, the findings during the training stage for the MLP (10:30:20:3) model are shown compared to the experimental data. Very high correlation coefficients (r^2^ = 1) were observed for all three output parameters considered.

During the validation stage, the neural model constructed for the third patient group received the 12 input data series reserved for this phase. The predictions from the constructed models were compared with the experimental data. The best results for the three output parameters were obtained with the MLP (10:30:20:3) model.

The excellent results obtained during the training stage for the highest performing MLP (10:30:20:3) model were confirmed during the validation stage.

According to the results shown in Figure 13, the probability of obtaining correct responses during validation was 67% (8 out of 12 predictions close to experimental values) for PSD, 83% (10 out of 12 predictions close to experimental values) for RISK, and 100% (12 out of 12 predictions close to experimental values) for C/D.

## 4. Discussions

Glaucoma is a challenging disease to understand, presenting difficulties in providing accurate and timely diagnosis and prognosis. Currently, no algorithm guarantees the set up of an optimal network, as optimizing the network architecture must fulfill multiple objectives. The “perfect” network must provide reliable predictions, avoid overtraining, ensure rapid convergence, minimize training time, offer better insights into the data-generating process (facilitating rule extraction), and reduce the time and cost involved in data collection and transformation. An optimal network should be large enough to learn the underlying function and sufficiently small to generalize effectively.

In this study, the obtained results demonstrated the potential of using these models to predict the progression of ocular hypertension (OHT) to glaucoma. The best performance for the group with untreated OHT was achieved using the MLP model (9:18:9:3). This neural model had a probability of providing correct responses during the validation phase of 75% (i.e., 9 responses close to the experimental values out of a possible 12) for the three output parameters: PSD, RISK, and C/D. For the group of patients with treated OHT, the most performant model was MLP (10:20:10:3). During the training phase, this model achieved correlation coefficients of 1 for RISK and C/D, and 0.9999 for PSD. The probability of generating correct responses during the validation phase with this model was 67% (8 out of 12) for PSD, 83% (10 out of 12) for RISK, and 75% (9 out of 12) for C/D. In the case of the control group, the best results were obtained with the MLP model (10:30:20:3). The correlation coefficient during training was 1 for all three output parameters. The probability of providing correct responses during the validation phase was 67% (8 out of 12) for PSD, 83% (10 out of 12) for RISK, and 100% (12 out of 12) for C/D. These results are similar to those reported in the literature. Therefore, the objective of this study is fulfilled through its ability to use these tools in predicting the progression of intraocular hypertension to glaucoma.

Various studies in the literature also demonstrate the successful use of these artificial intelligence tools in ophthalmology: for the assessment of the visual field, the optic nerve, and the retinal nerve fiber layer, thus providing greater accuracy in identifying glaucoma progression and retinal changes in diabetes [25,26,27,28,29,30]. The most recent reviews offer an analysis of all the studies in the literature that have employed artificial intelligence in ophthalmology—specifically in glaucoma—for predicting the progression of early changes that may occur in this disease, as well as in other conditions such as diabetic retinopathy, retinopathy of prematurity, and age-related macular degeneration [22,31].

In glaucoma screening, a key role is currently played by intraocular pressure measurement and fundus photography. Fundus photography is the fastest and simplest examination method to detect optic nerve damage caused by glaucoma. A recent study suggested that improving artificial intelligence (AI) programs could make glaucoma screening easier by analyzing color fundus photographs in a cost-effective manner. A study based on a large, labeled dataset of fundus photographs for glaucoma screening, which used AI, also showed that some specific features of glaucoma can be recognized and captured by AI [22,32]. Some recent studies have used deep learning models to estimate global and local visual field (VF) damage from raw OCT scans and quantified thickness measurements [33]. Another recent study employed additional AI tools, such as deep learning, with PyTorch. The PyTorch ANN models provided very good results in predicting visual field parameters (VFI, MD, and PSD) based on previous visit investigations [34]. In the literature, similar research highlighted the potential to predict 24-2 visual field defects up to 5.5 years into the future. This approach was complex, involving a large number of parameters and being based on deep learning neural networks that analyzed specific points in visual field images. Indeed, relatively small prediction errors were achieved; however, in our approach, the predicted visual field parameter values were lower than those reported by Wen et al. Moreover, this methodology involves using data from the patient’s medical record through a simpler method [35]. A study conducted by Thakur et al., 2023, discussed the application of deep learning models to predict the onset of glaucoma from fundus photographs, directly related to the task of predicting ocular hypertension that leads to glaucoma [36]. More recently, in 2023, Huang et al. described the GRAPE dataset, which includes multi-modal data such as longitudinal visual field (VF) measurements, fundus images, and clinical information. This dataset is crucial for training AI models to effectively predict the progression of ocular hypertension to glaucoma—similar to our study. In general, the GRAPE dataset could be used for prognostic prediction in glaucoma management and VF estimation, supporting the exploration of the structure–function relationship and advancing computer-assisted telemedicine in glaucoma care. The study advocated for the use of the GRAPE dataset to develop AI models for prognosis evaluation and VF estimation [37]. All these studies highlight the importance and usefulness of artificial intelligence tools in the diagnosis and prediction of glaucoma progression. By using diagnostic data as in our study, AI can help generate neural models that are applicable to large databases for predictive purposes.

### 4.1. Limitations and Challenges

This retrospective study involved a relatively small dataset. Obtaining patient consent for using AI models is critical. Moreover, applying AI to a particular ethnic group can be challenging if the AI algorithm is trained or tested on datasets with limited ethnic representation. Ethically, integrating AI into complex healthcare systems could disadvantage patients with multiple comorbidities by inadvertently reducing their priority.

### 4.2. Future Directions

Deep learning in retinal image analysis achieves excellent accuracy in the differential detection of retinal fluid types in the most common exudative macular diseases. Using OCT devices to monitor fluid volume evolution could allow for the optimization of anti-VEGF treatment. A deep learning network correctly indicates the need for intravitreal injections in 95% of cases [25,26].

Telemedicine—daily self-imaging conducted with home OCT at home for neovascular AMD [27]. Weekly image acquisition conducted using NVHO (Notal Vision Home OCT) over a 3-month period. The scans were uploaded to the cloud, analyzed using the Notal OCT Analyzer (NOA), assessed by human experts for the presence of fluid, and compared with in-office OCT scans. NVHO scans were analyzed with NOA and in-office OCT scans were assessed by human experts who agreed on fluid status in 96% of cases [27].

## 5. Conclusions

Glaucoma remains a complex disease to diagnose and monitor, and the use of artificial neural networks (ANNs) presents a promising alternative to traditional methods, especially in predicting progression from ocular hypertension (OHT) to glaucoma. The models developed in this study—particularly the Multilayer Perceptron (MLP) architectures—demonstrated good predictive capabilities across different patient subgroups, with validation phase accuracies ranging from 67% to 100% for key ophthalmologic parameters (PSD, RISK, and C/D). The main advantages of using these techniques in medical diagnostics include the ability to process large volumes of data, a low probability of overlooking relevant information, and a reduced time required to establish a diagnosis.

The robustness of these results was reinforced by rigorous statistical analyses. Specifically, the absence of missing data eliminated the need for imputation, thus avoiding potential bias. Given that the Shapiro–Wilk test indicated a non-normal distribution of the variables, a non-parametric Kruskal–Wallis one-way analysis of variance on ranks was applied. This revealed statistically significant differences in the median values across treatment groups: (H = 849.848, degrees of freedom = 11, *p* < 0.001)—Group I, (H = 865.380 degrees of freedom = 12, *p* < 0.001)—Group II, and (H = 1103.741, degrees of freedom = 12, *p* < 0.001)—Group III. Further, the Student–Newman–Keuls (SNK) test confirmed significant pairwise differences (*p* < 0.05), supporting the heterogeneity of the dataset and justifying the use of neural models capable of capturing complex, non-linear patterns.

The convergence behavior and predictive performance of the networks suggest that the high number of training epochs did not lead to overfitting, as monitored through MSE analysis and confirmed by consistent validation results. Moreover, the statistical heterogeneity of the dataset further justified extended training in order to generalize across diverse patient profiles.

## Figures and Tables

**Figure 1 life-15-00865-f001:**
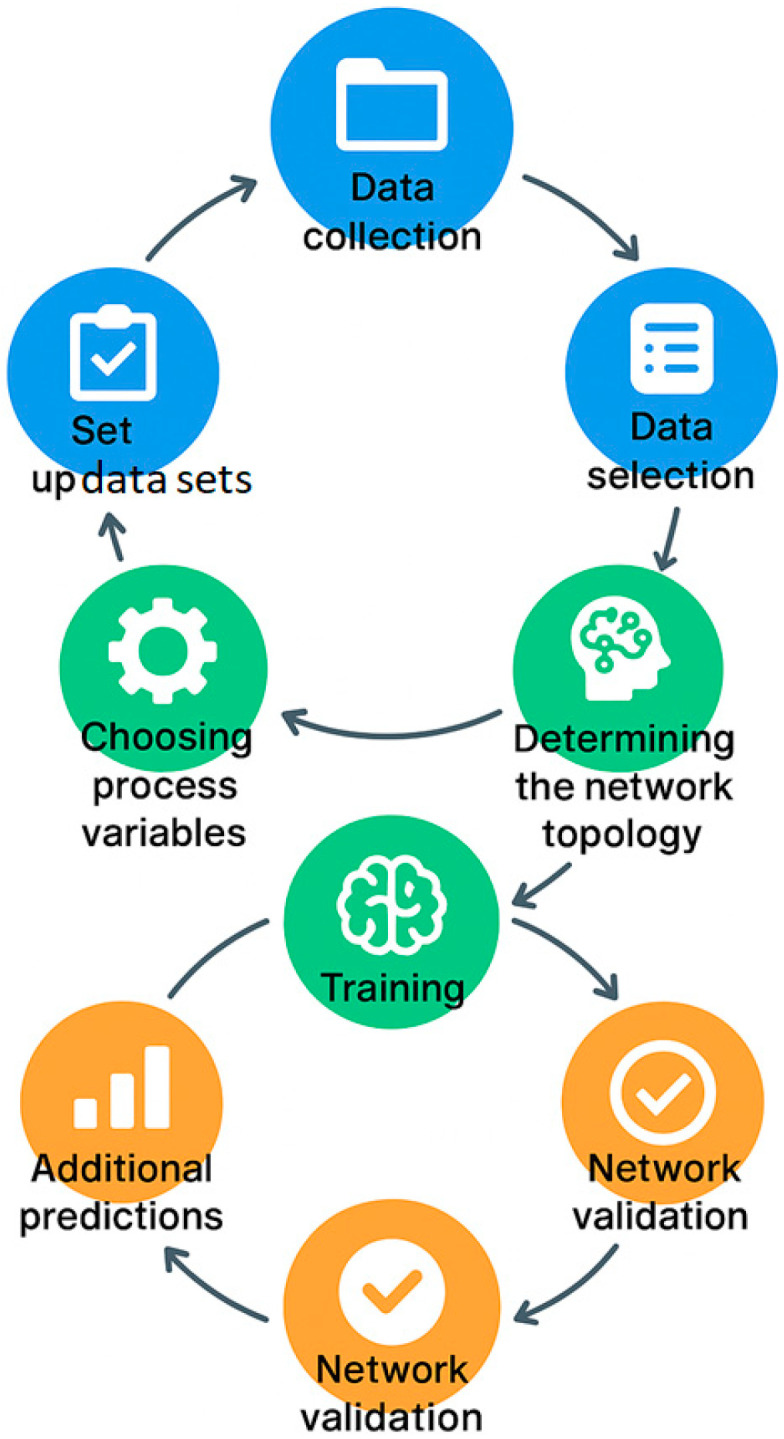
Stages of modeling with neural networks.

**Figure 2 life-15-00865-f002:**
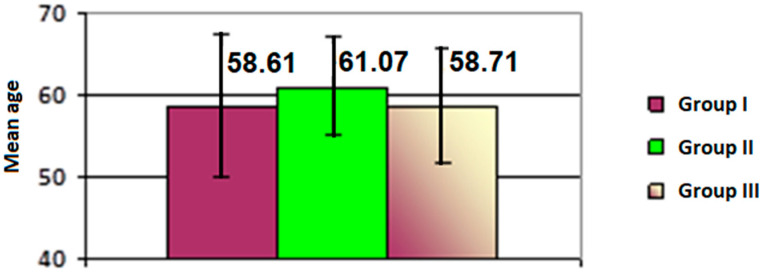
Mean age values across the study groups.

**Figure 3 life-15-00865-f003:**
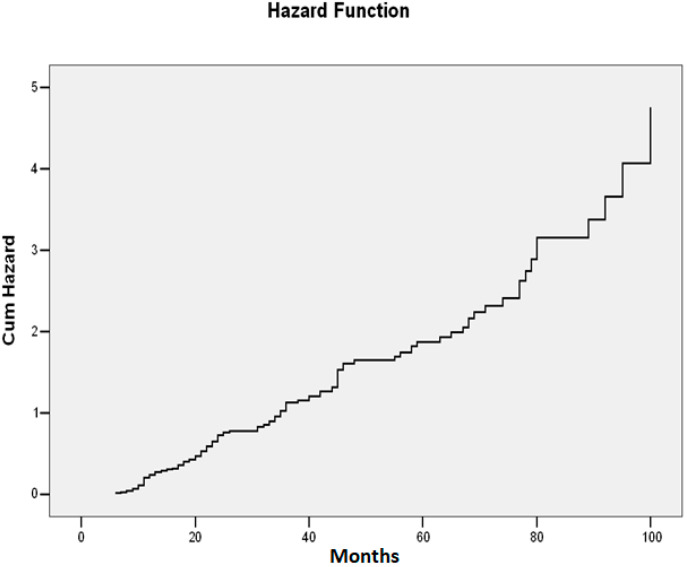
Five-year risk of conversion from intraocular hypertension (IOHT) to glaucoma.

**Figure 4 life-15-00865-f004:**
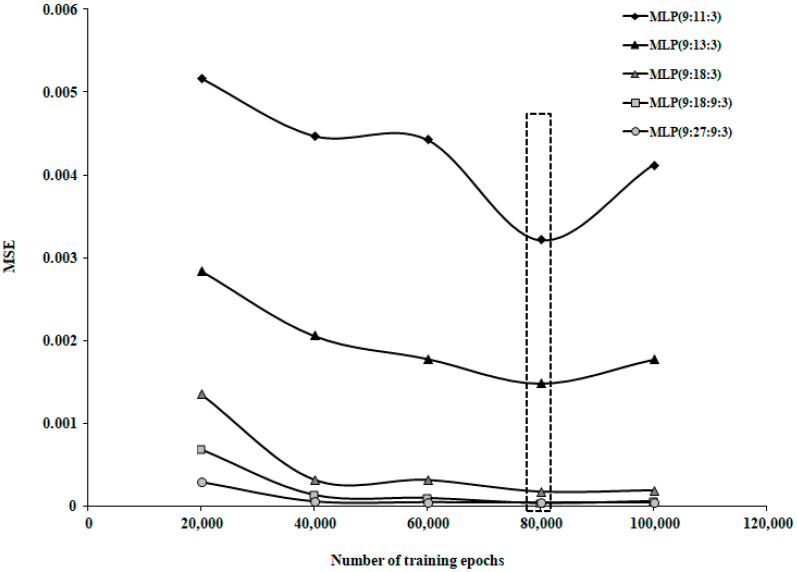
Variation in MSE with increasing number of training epochs for patient group 1.

**Figure 5 life-15-00865-f005:**
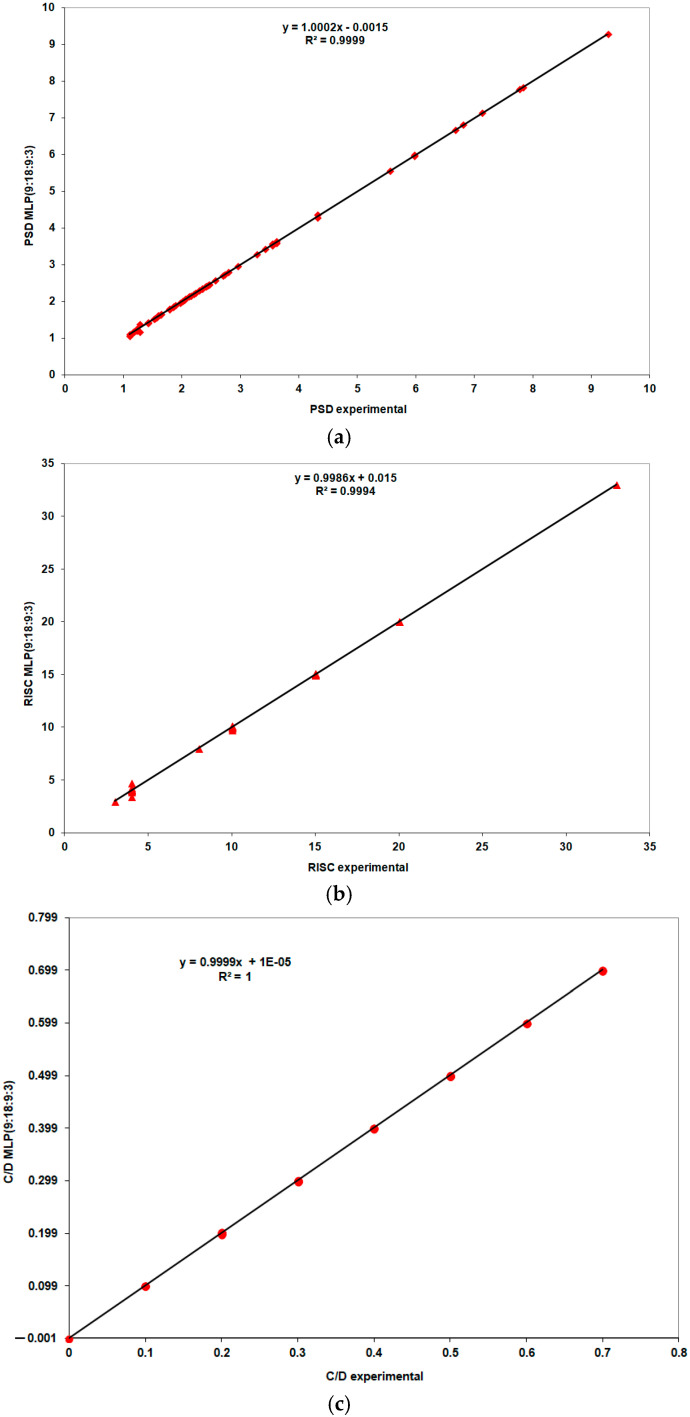
Performance evaluation of the MLP (9:18:9:3) model during the training stage: (**a**) PSD, (**b**) risk, (**c**) C/D.

**Figure 6 life-15-00865-f006:**
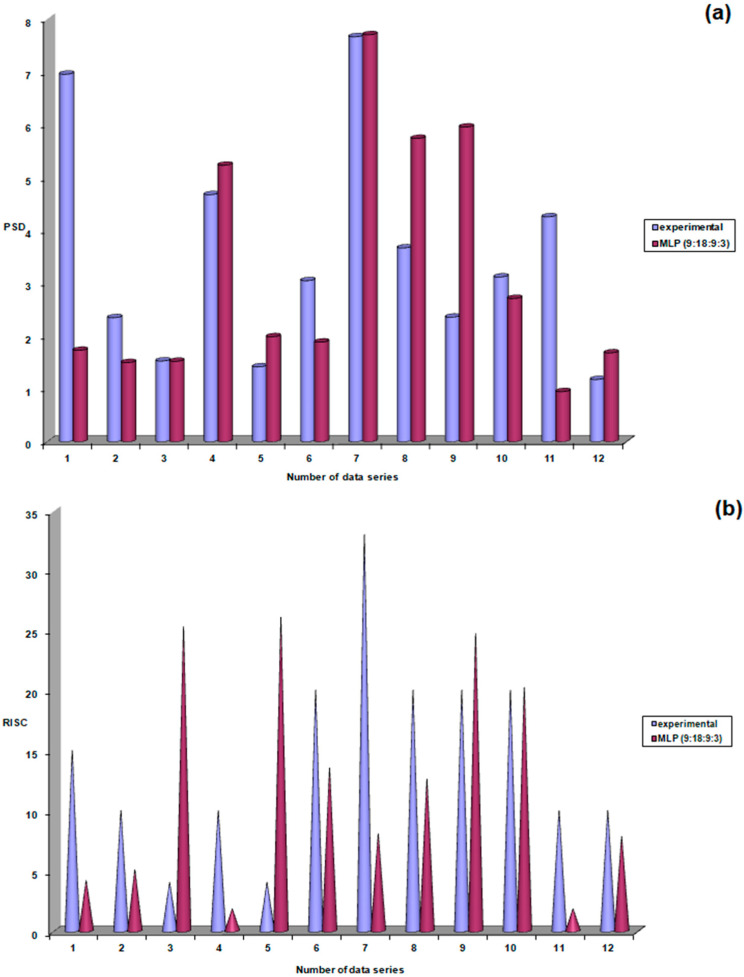
Performance evaluation of the MLP (9:18:9:3) model during the validation stage, (**a**) PSD (**b**) risk, (**c**) C/D.

**Figure 7 life-15-00865-f007:**
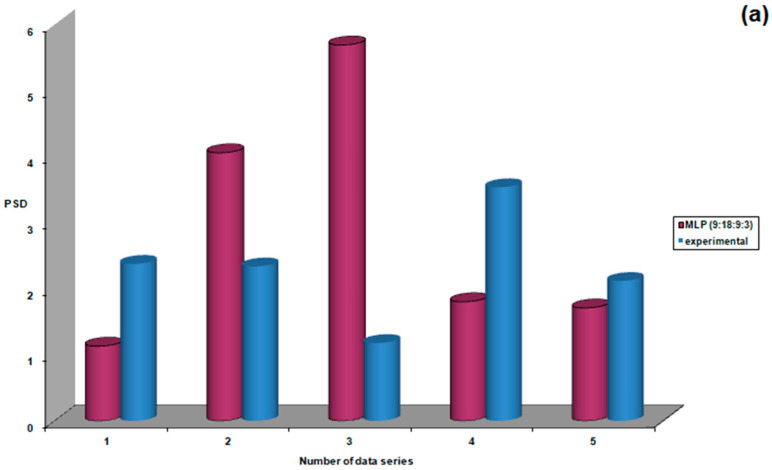
Performance evaluation of the MLP (9:18:9:3) model during the testing stage: (**a**) PSD, (**b**) risk, (**c**) C/D.

**Figure 8 life-15-00865-f008:**
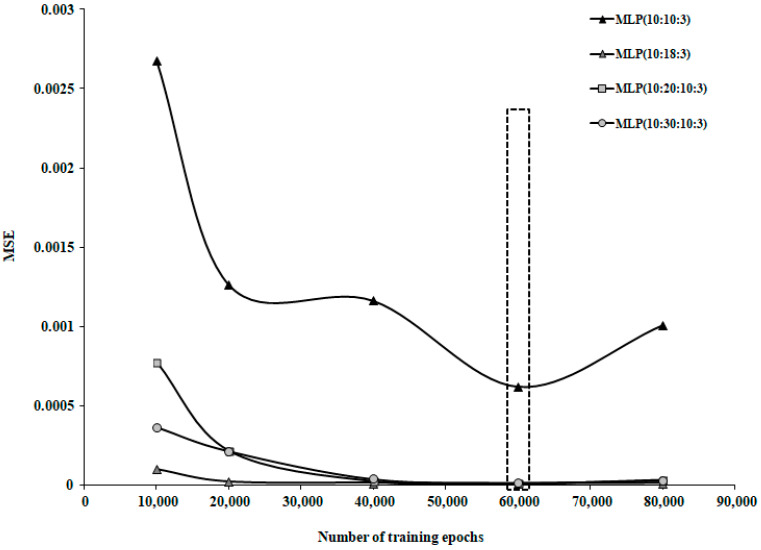
Variation in MSE with increasing number of training epochs for patient group 2.

**Figure 9 life-15-00865-f009:**
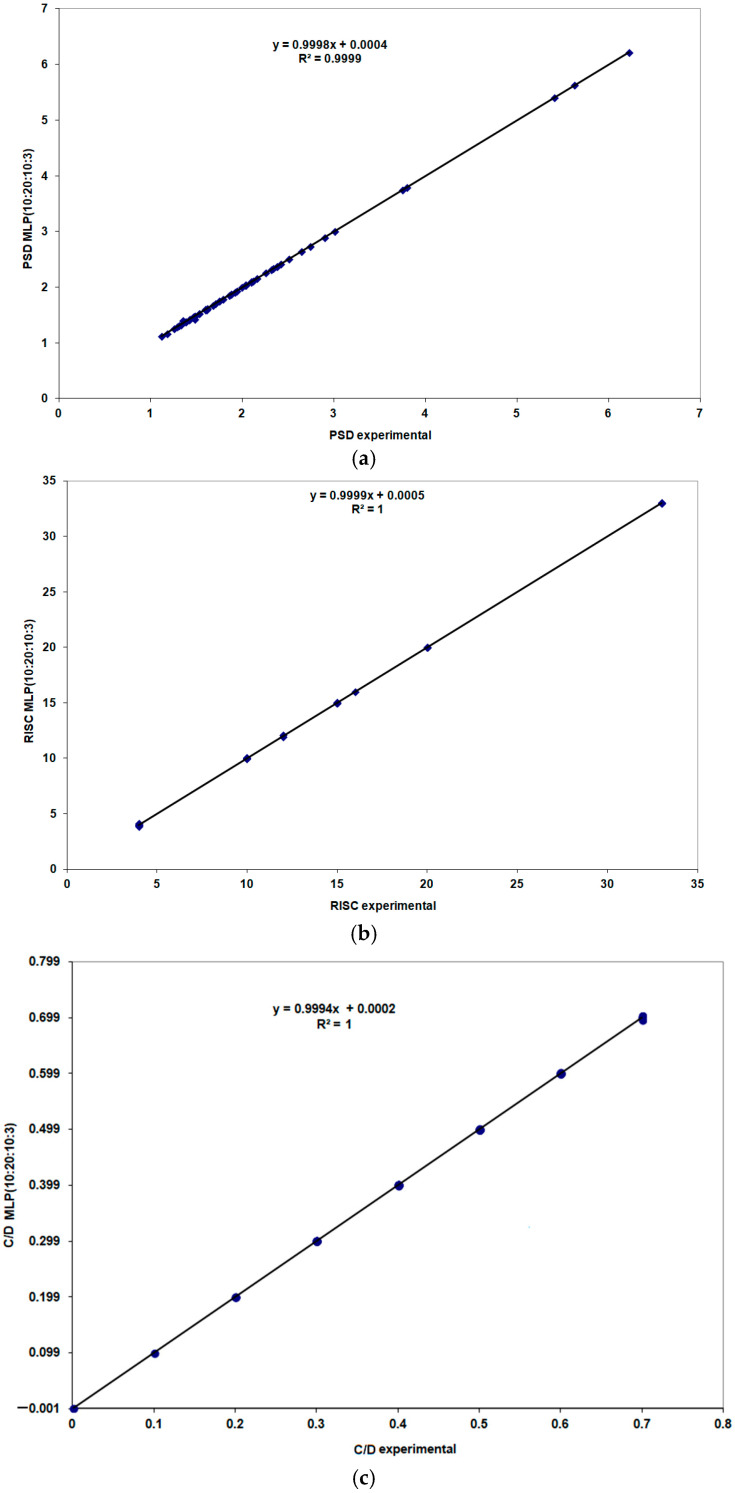
Performance evaluation of the MLP (10:20:10:3) model during the training stage: (**a**) PSD, (**b**) RISC and (**c**) C/D.

**Figure 10 life-15-00865-f010:**
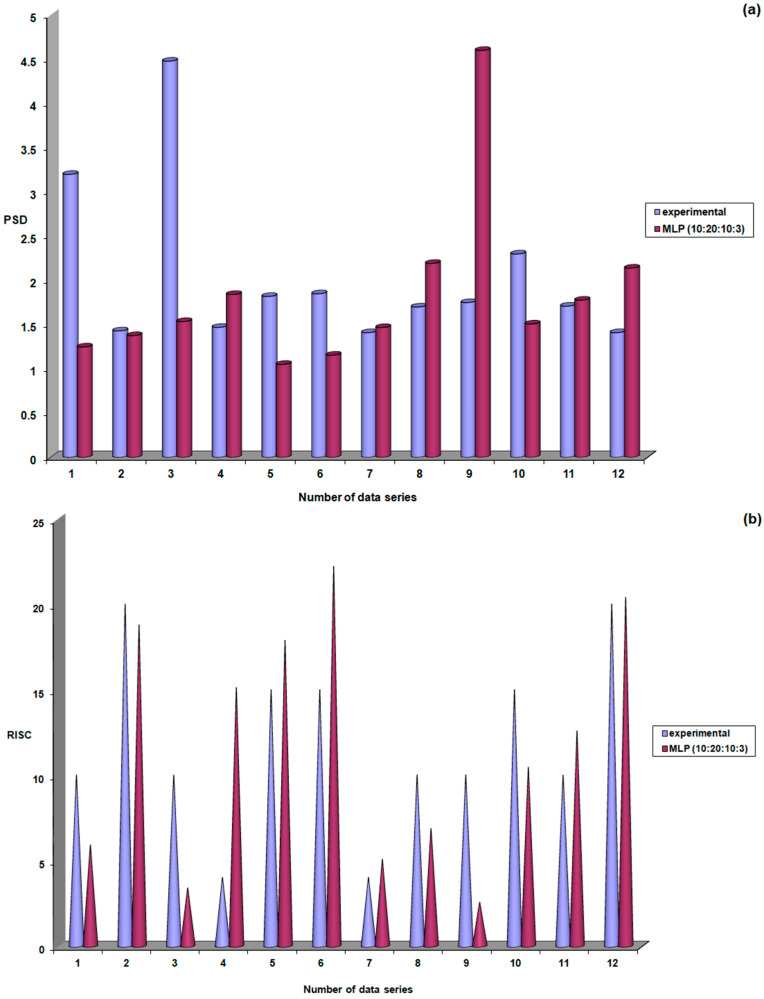
Performance evaluation of the MLP (10:20:10:3) model during the validation stage: (**a**) PSD; (**b**) RISC; (**c**) C/D.

**Figure 11 life-15-00865-f011:**
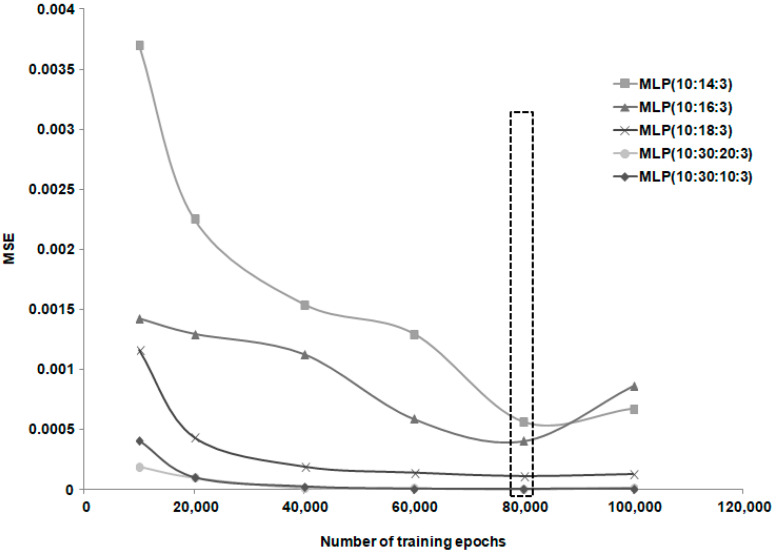
Variation in MSE as a function of the increasing number of training epochs for patient group 3.

**Figure 12 life-15-00865-f012:**
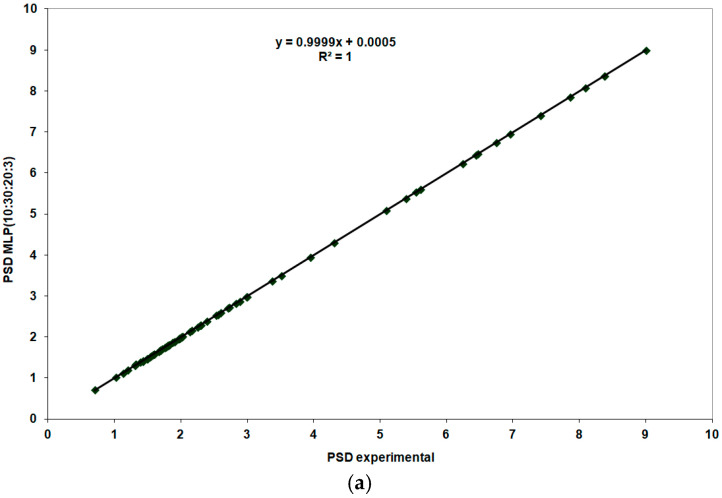
Performance evaluation of the MLP (10:30:20:3) model during the training stage: (**a**) PSD MLP (10:30:20:3) and MLP experimental; (**b**) RISC MLP (10:30:20:3) and RISC experimental; (**c**) C/D MLP (10:30:20:3) and C/D experimental.

**Figure 13 life-15-00865-f013:**
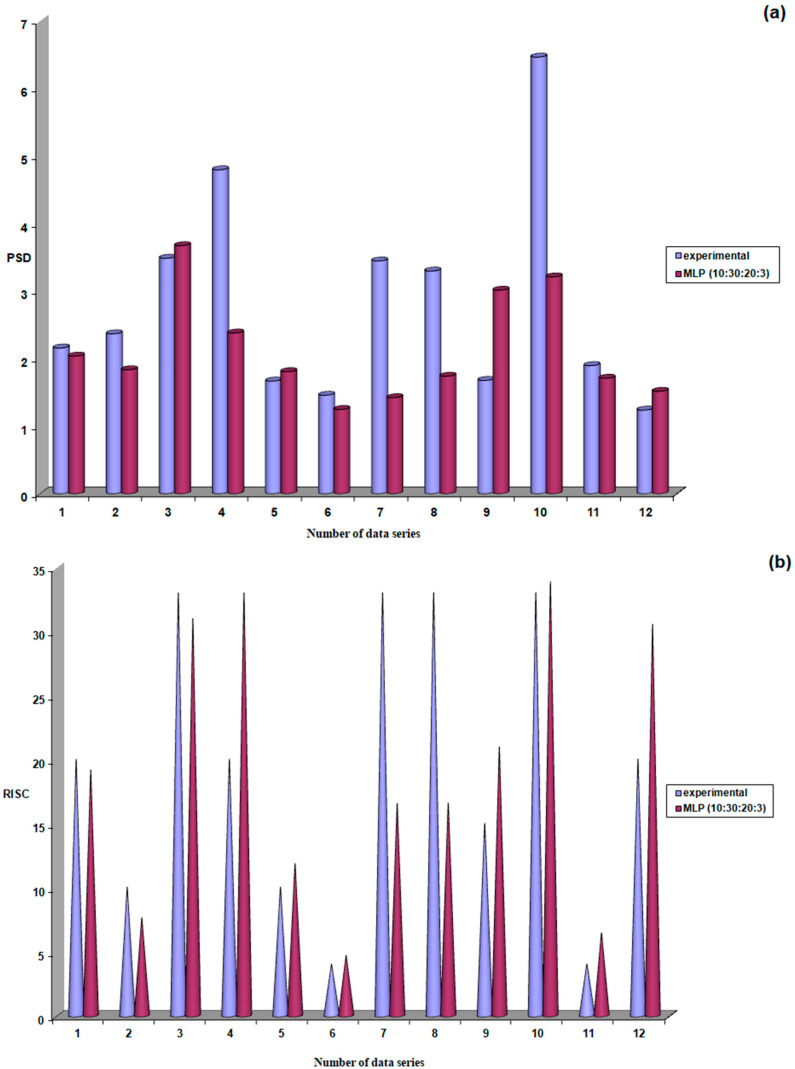
Performance evaluation of the MLP (10:30:20:3) model during the validation stage; (**a**) PSD; (**b**) RISC; (**c**) C/D.

**Table 1 life-15-00865-t001:** Statistical indicators of intraocular pressure (IOP, mmHg) compared across study groups.

Study Group	N	Mean	Standard Deviation	Standard Error	Confidence Interval 95%		Min	Max	F_ANOVA_ *p* Test
					−95%CI	+95%CI			
IOP maximum									
Group I	75	22.89	4.23	0.49	21.92	23.87	15	34	0.001
Group II	70	20.70	6.19	0.74	19.22	22.18	10	48	
Group III	89	19.71	3.48	0.37	18.97	20.44	10	33	
Total	234	21.03	4.84	0.32	20.40	21.65	10	48	
IOP minimum									
Group I	75	17.32	3.46	0.40	16.52	18.12	10	28	0.002
Group II	70	17.17	5.01	0.60	15.98	18.37	11	38	
Group III	89	15.39	3.24	0.34	14.71	16.08	9	30	
Total	234	16.54	4.00	0.26	16.03	17.06	9	38	

**Table 2 life-15-00865-t002:** Statistical indicators of the 5-year risk of conversion from intraocular hypertension (IOHT) to glaucoma, compared across study groups.

Study Group	N	Mean	Standard Deviation	Standard Error	Confidence Interval 95%		Min	Max	Test F_ANOVA_ *p*
					−95%CI	+95%CI			
Whole Group	234	13.66	8.57	0.56	12.55	14.76	3	33	-
Group I	75	16.83	9.80	1.04	14.77	18.89	4	33	0.001
Group II	70	12.21	7.86	0.94	10.34	14.09	4	33	
Group III	89	11.24	6.29	0.73	9.79	12.69	3	33	

**Table 3 life-15-00865-t003:** Linear regression model.

Model	R	R Square	Adjusted R Square	Std. Error of the Estimate	Change Statistics				
					R Square Change	F Change	df1	df2	Sig. F Change
1	0.106 (a)	0.011	0.007	8.542	0.011	2.637	1	232	0.106
2	0.309 (b)	0.096	0.088	8.187	0.084	21.537	1	231	0.001
3	0.313 (c)	0.098	0.086	8.193	0.003	0.675	1	230	0.412
4	0.314 (d)	0.099	0.083	8.208	0.001	0.169	1	229	0.681
5	0.492 (e)	0.242	0.225	7.544	0.143	43.055	1	228	0.001
6	0.492 (f)	0.242	0.222	7.561	0.000	0.012	1	227	0.914
7	0.614 (g)	0.377	0.358	6.868	0.135	49.073	1	226	0.001
8	0.768 (h)	0.590	0.575	5.585	0.213	116.768	1	225	0.001

a. Predictors: (Constant), Sex; b. Predictors: (Constant), Sex, age; c. Predictors: (Constant), Sex, age, systolic blood pressure; d. Predictors: (Constant), Sex, age, systolic blood pressure, diastolic blood pressure; e. Predictors: (Constant), Sex, age, systolic blood pressure (SBP), diastolic blood pressure (DBP), CCP (corneal central thickness); f. Predictors: (Constant), Sex, age, systolic blood pressure (SBP), diastolic blood pressure (DBP), CCP (corneal central thickness), IOP max; g. Predictors: (Constant), Sex, age, systolic blood pressure (SBP), diastolic blood pressure (DBP), CCP (corneal central thickness), IOP max, PSD; h. Predictors: (Constant), Sex, age, systolic blood pressure (SBP), diastolic blood pressure (DBP), CCP (corneal central thickness), IOP max, PSD, CD.

**Table 4 life-15-00865-t004:** Different MLP topologies tested for the first dataset.

No.	Network Topology	MSE	NMSE	r^2^	E_p_ (%)	Training Phase Length (Minutes)
1.	MLP(9:9:3)	0.008268	0.047467	0.974388	12.58	2.57
2.	MLP (9:11:3)	0.003215	0.018456	0.988966	8.39	4.36
3.	MLP (9:13:3)	0.001481	0.008502	0.994451	5.14	4.37
4.	MLP (9:15:3)	0.001156	0.006640	0.996061	4.87	4.36
5.	MLP (9:17:3)	0.000183	0.001048	0.999359	1.94	5.12
6.	MLP (9:18:3)	0.000175	0.001002	0.999431	1.87	4.23
7.	MLP (9:27:3)	0.000048	0.000273	0.999818	0.73	5.12
**8.**	**MLP (9:18:9:3)**	**0.000031**	**0.000176**	**0.999981**	**0.59**	**4.38**
9.	MLP (9:27:9:3)	0.000035	0.000185	0.999890	0.62	7.58

**Table 5 life-15-00865-t005:** Different MLP topologies tested for the second database.

No.	Network Topology	MSE	NMSE	r^2^	E_p_(%)	Training Phase Length (Minutes)
1.	MLP(10:10:3)	0.001849	0.009333	0.994498	5.26	2.23
2.	MLP (10:12:3)	0.000620	0.003130	0.998483	2.22	2.12
3.	MLP (10:14:3)	0.000309	0.001560	0.999258	1.77	1.77
4.	MLP (10:16:3)	0.000080	0.000405	0.999851	1.01	3.16
5.	MLP (10:18:3)	0.000009	0.000045	0.999977	0.21	3.19
6.	MLP (10:20:3)	0.000085	0.000428	0.999821	0.56	3.16
**7.**	**MLP (10:20:10:3)**	**0.000008**	**0.000040**	**0.999984**	**0.20**	**5.02**
8.	MLP (10:30:10:3)	0.000014	0.000069	0.999974	0.21	4.52

**Table 6 life-15-00865-t006:** Different MLP topologies tested for the third database.

No.	Network Topology	MSE	NMSE	r^2^	E_p_ (%)	Training Phase Length (Minutes)
1.	MLP(10:10:3)	0.005721	0.023217	0.986456	8.62	4.10
2.	MLP (10:12:3)	0.001885	0.007649	0.995992	5.54	4.04
3.	MLP (10:14:3)	0.000563	0.002283	0.998824	2.80	5.24
4.	MLP (10:16:3)	0.000406	0.001649	0.999310	2.22	3.98
5.	MLP (10:18:3)	0.000111	0.000452	0.999795	1.24	5.18
6.	MLP (10:20:3)	0.000105	0.000424	0.999821	1.03	5.38
7.	MLP (10:30:3)	0.000010	0.000041	0.999981	0.31	5.07
**8.**	**MLP (10:30:20:3)**	**0.000001**	**0.000004**	**0.999999**	**0.06**	**5.30**
9.	MLP (10:30:10:3)	0.000007	0.000029	0.999985	0.22	5.20

## Data Availability

The datasets used and analyzed in this study are available from the corresponding author on reasonable request.

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
