# Peer review of "The Role of Artificial Intelligence in Predicting the Progression of Intraocular Hypertension to Glaucoma"

_life, 2025, doi:10.3390/life15060865_

Round 1

Reviewer 1 Report

Comments and Suggestions for Authors

FOR THE INTRODUCTION:

  1. The introduction transitions to clinical tests without first detailing the pathophysiology cascade.
  2. You have invoked AI benefits before presenting any unmet analytical gaps in current prognostic scoring systems.
  3. Remove obstetric examples or justify their inclusion by analogy to longitudinal screening protocols.
  4. Lines 84-88 cite a 2021 DE-SVM study but provide no dataset size or public availability.
  5. Specify whether pachymetry values were corrected for corneal hysteresis interaction effects.
  6. The introduction fails to state the clinical decision point that the model will influence, such as treatment initiation criteria.

FOR THE RELATED LITERATURE:

7. Many references cluster before 2022 despite rapid field growth. Missing 2023-2025 hybrid transformer models skews the trend narrative.

8. Sleep apnea–glaucoma link is presented without causal pathways. Map how AI detects subtle nerve fiber changes earlier.

9. The section neglects cost-effectiveness literature. AI deployment depends on economic viability.

10. Citations 25-27 group oncology, obstetrics, and ophthalmology with no transition. Such abrupt theme jumps break conceptual coherence.

FOR THE METHODS:

11. Group allocation seems random but the method is not shown. Unclear sampling may hide selection bias.

12. Only one 75-25 split separates training, validation, and testing. Adopt k-fold cross-validation for robust error estimates.

13. Early stopping criteria are not described.

14. Momentum learning is named but the rate value is absent. Learning rate and momentum strongly affect stability. Supply exact α and β parameters.

15. Statistical tests assume normality without testing it. Shapiro–Wilk or Kolmogorov–Smirnov checks are absent.

FOR THE RESULTS & DISCUSSION

16. Update the results and discussions based on recommendations above.

17. The statistical tests assume normal data but the authors did not report any normality check. 

18. Confidence intervals are absent for key mean differences, so clinical magnitude is unclear.

19. Eighty-thousand epochs on 63 samples is extreme and risks memorization.

20. Momentum hyper-parameters are not reported, impeding exact replication.

21. Missing data handling is not described, so imputation bias is possible.

FOR THE CONCLUSIONS:

22. Update the conclusions based on my recommendations.

Author Response

Thanks for the comments and suggestions. I have attached the answers

kind regards

Reviewer 2 Report

Comments and Suggestions for Authors

The manuscript presents a research on application of MLP to train a predictive model of glaucoma disease progress. The theme is topical since nowadays AI applications in medical diagnostics increased.

The overall presentation is clear. However, several questions must be answered before paper acceptance:

  1. Why for each one of the three patients groups having different stages of glaucoma a separate MLP model is trained? If the model intends to predict the future disease progress it should be valid though all the cases from early detected through progressive glaucoma stages.
  2. How much of previous authors' own works [19, 20] is included in the current manuscript? I se that some figures, e.g. Figure 1 from [20] are exactly the same. It is important to decrease self-plagiarism!
  3. Additionally some technical issues must be resolved like: many figures have left text in Romanian language; some figure's legends also contain Romanian words.

Author Response

(The authors gave the same response as above.)

Round 2

Reviewer 1 Report

Comments and Suggestions for Authors

1. I am fine with the revisions, no more questions.

Author Response

Thank you so much for all sugestions

kind regards

Nicoleta Anton

Reviewer 2 Report

Comments and Suggestions for Authors

Please clarify in more details why (according to your answer) "Separate groups were taken because in all groups there were different values of the entry criteria"? In my opinion even if the three groups have different values of input variables, this should not be a problem to train more general model. I still do not accept such an approach. It allows to achieve high accuracy of three models but for sure they are not universal.

On most of figures are still left Romanian notions on x axes, e.g. Luni on Fig. 3, Numarul de epoci de antrenare on Fig. 4, Nr de serii de date on Fig. 6, 7, 10 and 13.

Author Response

(The authors gave the same response as above.)

Round 3

Reviewer 2 Report

Comments and Suggestions for Authors

Dear Authors, please include the explanations from your response letter into the manuscript text.

I am still unconvinced in your idea. Please explain why you do not use initial tension values as well as treatment/no treatment case as input variable to your models. This will allow to train much more realistic model that would be able to predict progression of glaucoma.

Please discuss the reasons why you decided to train three but not a single model in details in the manuscript to make your research more convincing!

Round 3

I am still unconvinced in your idea. Please explain why you do not use initial tension values as well as treatment/no treatment case as input variable to your models. This will allow to train much more realistic model that would be able to predict progression of glaucoma.

Please discuss the reasons why you decided to train three but not a single model in details in the manuscript to make your research more convincing!

Authors’ Response:

We have improved the quality of the article according to what you suggested, for which we thank you, as well for your professional evaluation, time and efort! Your suggestions improved our manuscript.

.

Before making the prediction with neural networks, we analyzed the database. Based on the analysis of the database, where we identified statistically significant correlations, we included input and output parameters in the study. Another important landmark was The linear regression model 7 shows that in 59% of cases, individual values ​​of blood pressure, central corneal thickness, intraocular pressure, pattern deviation, and the cup-to-disc ratio are significant predictors of an increased risk score for glaucoma pro-gression (p = 0.001). The 5-year risk of conversion from intraocular hypertension (IOHT) to glaucoma was significantly higher in untreated IOHT patients (16.83%) compared to those with treated IOHT (12.21%) and treated POAG (control group) (11.24%), with a statistically significant difference.

Thus, a total of nine input parameters were considered: age, sex, months, diagnosis group (DG), systolic blood pressure, diastolic blood pressure, pachymetry, maximum intraocular pressure (IOP max), and minimum intraocular pressure (IOP min). The three output parameters were: Pattern Standard Deviation (PSD), risk score (RISC), and cup-to-disc ratio (C/D). These parameters were used in all three groups.

Thus, we tried to obtain the best networks for each group that would make the best predictions about treated IOP in glaucoma versus untreated IOP. The presence of intraocular hypertension treatment may positively influence the delay of intraocular hypertension progression in glaucoma.

I also added it to the text in yellow

Thank you so much for your sugestions

Kind regards

Nicoleta Anton
